# Super-Resolution STED Microscopy-Based Mobility Studies of the Viral Env Protein at HIV-1 Assembly Sites of Fully Infected T-Cells

**DOI:** 10.3390/v13040608

**Published:** 2021-04-02

**Authors:** Jakub Chojnacki, Christian Eggeling

**Affiliations:** 1MRC Human Immunology Unit, Weatherall Institute of Molecular Medicine, University of Oxford, Oxford OX3 9DS, UK; christian.eggeling@uni-jena.de; 2IrsiCaixa AIDS Research Institute, University Hospital Germans Trias i Pujol, Ctra. de Canyet s/n, Badalona, 08916 Barcelona, Spain; 3Institute of Applied Optics and Biophysics, Friedrich-Schiller-University Jena, Max-Wien Platz 4, 07743 Jena, Germany; 4Leibniz Institute of Photonic Technology e.V., Albert-Einstein-Straße 9, 07745 Jena, Germany

**Keywords:** HIV-1, virus assembly, super-resolution, microscopy, FCS, STED-FCS

## Abstract

The ongoing threat of human immunodeficiency virus (HIV-1) requires continued, detailed investigations of its replication cycle, especially when combined with the most physiologically relevant, fully infectious model systems. Here, we demonstrate the application of the combination of stimulated emission depletion (STED) super-resolution microscopy with beam-scanning fluorescence correlation spectroscopy (sSTED-FCS) as a powerful tool for the interrogation of the molecular dynamics of HIV-1 virus assembly on the cell plasma membrane in the context of a fully infectious virus. In this process, HIV-1 envelope glycoprotein (Env) becomes incorporated into the assembling virus by interacting with the nascent Gag structural protein lattice. Molecular dynamics measurements at these distinct cell surface sites require a guiding strategy, for which we have used a two-colour implementation of sSTED-FCS to simultaneously target individual HIV-1 assembly sites via the aggregated Gag signal. We then compare the molecular mobility of Env proteins at the inside and outside of the virus assembly area. Env mobility was shown to be highly reduced at the assembly sites, highlighting the distinct trapping of Env as well as the usefulness of our methodological approach to study the molecular mobility of specifically targeted sites at the plasma membrane, even under high-biosafety conditions.

## 1. Introduction

Plasma membrane lipid and protein organisation modulates a wide range of biological processes [1]. The details of this organisation can be probed in a live cell context through interrogation of the diffusion characteristics of molecules using fluorescent probe-based techniques such as fluorescence correlation spectroscopy (FCS) [2]. FCS allows for the determination of the average transit time (τ_D_) of a tagged molecule through a confocal microscope excitation volume of known size, obtaining the molecular diffusion coefficient (D) of the observed molecule. FCS has previously been used to study changes in membrane viscosity [3] as well as hindered diffusion in cells due to molecular interactions [4].

Although FCS is characterised by high temporal resolution, in its classical form, it is unsuitable for observation of processes such as endocytosis, virus assembly and entry, as well as cell surface receptor clustering, as these occur at spatial scales below the resolution (or diffraction) limit of conventional light microscopy. This limitation can be remedied by the combination of FCS recordings with stimulated emission depletion (STED) super-resolution microscopy (STED-FCS), which allows for observations at sub-diffraction spatial scales (ranging from 240 nm to approx. 40 nm) [5]. Another limitation of FCS lies in the fact that it allows for the acquisition of molecular dynamics in only a single fixed observation spot, which limits its usability in the study of spatially heterogeneous cell membranes. This can be addressed by adding a spatial scanning component to FCS where data can be acquired simultaneously at various points along a scanned line, i.e., scanning FCS [6]. By combining the above modalities, scanning STED-FCS (sSTED-FCS) becomes a powerful tool for the interrogation of local heterogeneities in cell membranes on sub-diffraction scales [7,8,9].

To effectively study the membrane behaviour in the context of cellular processes such as endocytosis or virus entry assembly, it is important to set the FCS scanning line into the precise area of interest. This can be achieved by fluorescently labelling another molecule which participates in the process and using it a guide for FCS data acquisition. Such a two-colour sSTED-FCS approach has recently been utilised to study the behaviour of fluorescent lipid analogues inside and outside of HIV-1 assembly sites, using the structural protein Gag to guide FCS observation points to the relevant areas [10].

The HIV-1 assembly site is a sub-diffraction-sized (≈140 nm) area of Gag protein multimerisation which leads to the release of the virus particle from the plasma membrane of the infected cell. The Gag protein, a key player in this process, is a major structural protein in the virus and mediates interactions between the assembling virus particle and plasma membrane lipids (Figure 1). Its four domains (MA, CA, NC and p6) are responsible for targeting Gag towards the cell plasma membrane (MA), mediation of Gag–Gag multimerisation interactions (CA), RNA encapsidation (NC) and the recruitment of cell factors required for HIV-1 particle budding from the host cell plasma membrane (p6) [11]. Env is another key HIV‑1 protein responsible for virus attachment and fusion with the target cells. During HIV-1 assembly, only 7–10 copies of this trimeric glycoprotein become incorporated into the virus particle in the process regulated by interactions between the MA domain of Gag and the Env C-terminal tail [12,13,14]. 

While previous studies have shown Env trapping and enrichment at HIV-1 assembly sites, these studies have either used incomplete replication-incompetent virus models [15,16] or orthologous Env proteins [17]. Here, we applied the two‑colour sSTED-FCS approach to study the dynamic behaviour of a virus protein, Env, inside and outside HIV-1 assembly sites in the context of cells infected by fully infectious NL4.3 HIV-1, using a microscope placed in a biosafety level 3 environment. We demonstrate that this protein exhibits highly reduced mobility at the virus assembly sites identified by the fluorescent Gag clusters, compared to the areas immediately outside. This study further highlights the usefulness of the two-colour sSTED-FCS approach in the study of the heterogeneous behaviour of molecules at discrete plasma membrane sites.

## 2. Materials and Methods

### 2.1. Antibodies, Lipids and Fluorescent Lipid Analogues

Human anti-gp120 monoclonal antibody 2G12 was purchased from Polymun Scientific, Austria. 2G12 Fab fragments were generated using the Pierce Fab Micro Preparation kit (Thermo Fisher Scientific Inc., Waltham, MA, USA) according to the manufacturer’s instructions. Anti-human IgG Fab fragments (Jackson ImmunoResearch Europe, Ely, UK) were coupled to Abberior STAR RED (KK114) dye (Abberior GmbH, Göttingen, Germany) via NHS-ester chemistry according to the dye manufacturer’s instructions. In addition, 1,2-dioleoyl-sn-glycero-3-phosphocholine (DOPC) was purchased from Avanti Polar Lipids, Alabaster, AL, USA. Abberior STAR RED (KK114)–1,2-dihexadecanoyl-sn-glycero-3-phosphoethanolamine (KK114-DPPE) was purchased from Abberior GmbH, Göttingen, Germany [10]. 

### 2.2. Plasmids

Plasmid-expressing, fully infectious T-cell tropic NL4.3 HIV-1 Gag.iGFP (previously described in [18]) was a gift from Benjamin Chen [10].

### 2.3. Cell Culture

The 293T cells (ATCC CRL-3216) were grown in Dulbecco’s modified Eagle’s medium (Sigma-Aldrich, St. Louis, MO, USA), supplemented with 10% foetal calf serum, 100 U/mL penicillin–streptomycin and 20 mM HEPES pH 7.4. Cells were maintained at 37 °C, 5% CO_2_. Jurkat T-cells (ATCC TIB-152) were grown in RPMI 1640 (Sigma-Aldrich) supplemented with L-Glutamine, 10% foetal calf serum, 100 U/mL penicillin–streptomycin and 20 mM HEPES (pH 7.4). Cells were maintained at 37 °C, 5% CO_2_**.**

### 2.4. NL4.3 HIV-1 Gag.iGFP Particle Generation

Fully infectious NL4.3 HIV-1 Gag.iGFP particles were prepared from the 293 T-cell culture supernatants. Cells were transfected using polyethyleneimine (PEI) with 15 mg of pNL4.3 HIV-1 Gag.iGFP plasmid. Tissue culture supernatants were harvested 48 h after transfection and particles were concentrated using a Lenti-X Concentrator reagent (Clontech, Mountain View, CA, USA) according to the manufacturer’s instructions. Concentrated particles were snap-frozen and stored in aliquots at −80 °C [10].

### 2.5. Jurkat T-Cell Infection

Jurkat T-cells were infected by incubation of 1 million cells with HIV-1 NL4.3 Gag.iGFP particles in 50 mL of RPMI medium for 1 h at 37 °C. Cells were washed three times in RPMI medium and cultured for 72 h at 2 million cells/mL to achieve a 5–10% infection rate with progeny virus production [10].

### 2.6. Microscope Setup

sSTED-FCS measurements were performed on a Abberior Instrument Expert Line STED super-resolution microscope (Abberior Instruments GmbH, Göttingen, Germany) placed in a biosafety level 3 environment, using 485- and 640-nm pulsed excitation laser sources and a pulsed STED laser operating at 775 nm and an 80-MHz repetition rate. The fluorescence excitation and collection were performed using a 100×/1.40 numerical aperture (NA) UPlanSApo oil immersion objective (Olympus, Southend-on-Sea, UK). All acquisition operations were controlled by Imspector software (Abberior Instruments GmbH, Germany). Point FCS data for the estimation of observation spot size were recorded using a hardware correlator (Flex02-08D, correlator.com, Newark, NJ, USA operated by the company’s software) [10].

### 2.7. Excitation Spot Size Estimation via Supported Lipid Bilayers (SLBs)

SLBs were used for the estimation of the excitation spot size, as described previously [19]. SLBs were created by spin-coating a pre-cleaned coverslip with a solution of 1 mg/mL KK114-DPPE and DOPC at 1:2000 ratio in CHCl3/MeOH. Coverslips were pre-cleaned by piranha solution (3:1 sulfuric acid and hydrogen peroxide). The lipid bilayer was formed by rehydrating with SLB buffer containing 10 mM HEPES and 150 mM NaCl (pH 7.4) [10].

### 2.8. Sample Preparation

For protein mobility measurements inside and outside Gag assembly sites, 3 × 10^5^ Jurkat T-cells were suspended in in 200 mL Leibovitz’s L-15 medium (Sigma-Aldrich) 72 h post-infection with NL4.3 HIV-1 Gag.iGPF. Infected cells were stained for Env in suspension at 16 °C by incubation with 2G12 Fab fragments and anti-human Abberior STAR RED (KK114) conjugated Fab fragments for 1 h each in 0.5% BSA/L-15 medium. Cells were then washed three times in L-15 medium, resuspended in 250 mL of L-15 medium and adhered to solution poly-L-lysine (0.1 mg/mL) (Sigma-Aldrich)-coated glass surface of eight-well Ibidi m-Slides (Ibidi, Gräfelfing, Germany) at 37 °C 30 min prior to microscopy measurements.

### 2.9. Linear sSTED-FCS Signal Acquisition

sSTED-FCS data of Env mobility inside and outside Gag assembly sites in Jurkat T-cells were acquired at 37 °C using fluorescent Gag signal as a guide. Env signal intensity fluctuation carpets were recorded using Imspector software with the following parameters: line scan frequency, 1.92 kHz; scan line length, 1 µm; pixel dwell time, 10 µs; total measurement time, 10 s; pixel size, 50 nm per pixel, excitation power (back aperture), 10 µW at 640 nm; and observation spot diameter, 100 nm (full width at half maximum) FWHM (as determined by SLB-based excitation spot size estimation). The scan frequency selected for these experiments was set sufficiently high to enable the extraction of protein diffusion dynamics in the plasma membrane.

### 2.10. FCS Curve Autocorrelation and Fitting

The FoCuS-scan software [20] was used to autocorrelate the scanning STED-FCS data and to fit them with a classical 2D diffusion model:(1)$$G_{N}\left(\tau \right)=G\left(\infty \right)+G_{N}\left(0\right){\left[1+{\left(\frac{\tau }{t_{xy}}\right)}^{\alpha }\right]}^{-1}$$
where *G_N_(τ)* is the correlation function at time lag *τ*, G(∞) the offset, *G_N_*(0) the amplitude, *t_xy_* the average lateral transit time through the observation spot, and α is an anomaly factor which takes into account possible deviations (due to, for example, photobleaching or high curvature) from the assumed purely lateral 2D diffusion (α = 1 ideal case).

Values of the molecular diffusion coefficient (*D*) were calculated from *t_xy_*:(2)$$D=FWHM^{2}/8\ln\left(2\right)t_{xy}$$
where *FWHM* represents the usual full-width half maximum of the observation spot (here, 100 nm).

### 2.11. Statistical Analysis

Due to the non-Gaussian nature of the diffusion coefficient data distribution, Wilcoxon test was used, with a result *p* < 0.05 considered statistically significant. Statistical tests were performed using Graphpad Prism software. Power calculations confirmed that for chosen sample sizes, the power of a two-sided hypothesis test at 0.05 significance was over 90%. In all compared groups, interquartile range (IQR) served as an estimate of variation. IQRs were similar in all compared groups [10].

## 3. Results and Discussion

### 3.1. HIV-1 Env Mobility Can Be Determined Inside and Outside HIV-1 Assembly Sites by sSTED-FCS

To analyse the behaviour of Env at individual HIV-1 assembly sites via two-colour sSTED-FCS, we utilised a STED microscope placed in a biosafety level 3 environment and a replication-competent NL4.3 virus to infect Jurkat CD4+ T-cell lymphocytes. While retaining full infectivity, this virus expresses green fluorescence protein inserted between the MA and CA domains of Gag (Gag.iGFP) [18], which allows for the identification of punctate virus assembly sites on the plasma membrane of the infected cell. This virus model has been previously shown to support the production of individual virus assembly sites [10]. Env molecules on the surfaces of infected cells were visualised by immunofluorescence using anti-Env 2G12 Fab fragments prior to adhering cells to poly-L-Lysine-coated coverslips (Figure 2a).

The acquisition of the fluorescently tagged Env inside and outside assembly sites was guided by the Gag.iGFP signal, which was used to align the STED-FCS scan line (Figure 2b). Newly appearing (post cell adherence) small and immobile Gag.iGPF clusters were selected for subsequent analysis. After the alignment with the Gag.iGPF cluster, the microscope was switched from imaging (xy axis) mode to scanning line (xt axis) mode. Temporal fluorescence intensity fluctuations were then recorded at each pixel position along the multiple scanned lines (denoted “intensity fluctuation carpet”), including areas with and without Gag.iGPF fluorescence, which correspond to areas inside and outside the virus assembly site. We first quickly acquired a fluorescence intensity fluctuation carpet for the Gag.iGFP signal in confocal mode (Figure 2c), in order to confirm that (1) the scanned line was correctly aligned with the virus assembly site, and (2) that the observed virus assembly site was sufficiently immobile to allow for observations unbiased by, for example, drifting of the Gag.iGFP signal. Subsequently, we acquired the intensity fluctuation carpet for the Env signal in STED mode (100 nm observation spot size) (Figure 2d), finally followed by the acquisition of another fluorescence intensity fluctuation carpet of Gag signal in confocal mode to ensure negligible drift of the observed virus assembly site (Figure 2e).

The parameters for the acquisition of Env fluorescence fluctuation data were selected to provide accurate data with minimal bias with: (1) 100 nm observation spot size to observe Env mobility only in the area within the HIV-1 assembly site, thus allowing us to discriminate between Env behaviour inside and outside assembly sites, and (2) 1.92 kHz linear scanning, which provides sufficient temporal resolution to determine the mobility of proteins in the plasma membrane while minimising excitation and STED laser exposure, thus reducing phototoxicity [7,21]. Fluorescence intensity fluctuation data for each point of the scanned line were autocorrelated and fitted with a generic two-dimensional (2D) diffusion model to obtain the average transit times and diffusion coefficient (D) of Env molecules for spots inside and outside the HIV-1 assembly site (Figure 2f).

### 3.2. Env Becomes Trapped at HIV-1 Assembly Sites in Infected CD4+ T-Cells

Two-colour sSTED-FCS allowed for a successful observation of Env mobility within (red boxes) and outside (green boxes) HIV-1 assembly sites on the surfaces of infected Jurkat T-cells (Figure 3a). Results indicated that the median diffusion coefficient of Env molecules within the assembly site is very highly reduced (D_median_ = 0.004 µm^2^/s, interquartile range (IQR) = 0.001 µm^2^/s) compared to the mobility outside the assembly site (D_median_ = 0.096 µm^2^/s, IQR = 0.089 µm^2^/s). We observed no such confinement for experiments of phosphatidylethanolamine (PE) lipid mobility inside the virus assembly site (Figure 3b), performed as a part of our related lipid mobility study [10]. The outside Env mobility was similar to the result obtained previously [22] for sSTED-FCS measurements of Env plasma membrane mobility in HeLa cells transfected with Env-expressing construct (D_median_ = 0.074 µm^2^/s, IQR = 0.054 µm^2^/s). The direct relationship between the diffusion coefficient and spot size diameter is consistent with trapped diffusion behaviour that was previously reported for cell surface Env [22]. We have to note that the previously observed lower values of the Env diffusion coefficient in transfected HeLa cells were a result of performing this experiment with a STED observation spot of a smaller size—55 nm instead of 100 nm in diameter, as in this study—which was only possible due to the high signal-to-noise ratio of the Env signal in the transfected cell model.

The values of the diffusion coefficient of Env at the virus assembly sites (D = 0.002–0.004 µm^2^/s) were similar to those of Env on individual cell-free HIV-1 particles [22], also determined by sSTED-FCS, which indicates strong trapping of HIV‑1 Env molecules at the early stages of the virus assembly prior to virus budding. From our measurement protocol, we can exclude erroneous observation of slow diffusion of already budded particles, as these would be characterised by a drift in the Gag signal. Moreover, in our previous two-colour sSTED-FCS studies of lipid mobility at HIV-1 assembly sites, we observed no site-specific trapping of the independent immune receptor MHC-I protein (see [10], Supplementary Information), unlike in fully budded HIV-1 particles [22], thus indicating that observed sites of Gag clustering do not represent fully budded virus particles. The observed trapping of Env during fully infectious HIV-1 assembly is in agreement with live-cell observations of the accumulation of murine leukaemia virus Env (MLV-Env) concurrent with the formation of Gag clusters [17], as well as a recent study of Env mobility at HIV-1 assembly sites via single particle tracking [23]. Env trapping at the bud neck was also observed in studies of HIV-1 assembly sites utilising a Gag model lacking a PTAP motif in the p6 Gag domain to inhibit the virus release [15,16]. Thus, our results are in line with the previous observations where HIV-1 Env is trapped at the virus assembly sites via interaction with the nascent Gag lattice via the Env C-terminal tail [15].

The analysis of Env mobility outside virus assembly sites was challenging due to the low Env signal in these areas. The amplitude of FCS autocorrelation curves indicated a lower amount of fluorescent Env molecules in the observation volume outside the assembly site (*N* = 1–2) when compared to inside the assembly sites (*N* = 7–10). This difference can also be seen in the images of the virus assembly sites (Figure 2a,b). This comparative lack of Env molecules outside of virus assembly areas in infected T-cells is consistent with the previously proposed mechanism where the endocytosis and intracellular retention of Env restrict its cell surface levels [16] and lead to the incorporation of a low number of Env molecules (7–10) per virus particle [12,13].

Our present study highlights the usefulness of the two-colour sSTED-FCS approach in the observation of plasma membrane heterogeneities at specifically targeted sites. Unlike in camera-based methods such as Total Internal Reflection Fluorescence Microscopy (TIRFM) [17], the presented approach does not allow for the continuous monitoring of HIV-1 assembly sites over an extended period of time, due to the movements of the cell and assembly sites. Nonetheless, the sSTED-FCS approach enables measurement of the average Env mobility in the time window between the initial individual Gag cluster formation and virus budding. This shortcoming can be mediated in future studies by implementing single Gag cluster tracking and automated sSTED-FCS data acquisition at selected points and times into the acquisition pipeline.

The data in this study provide evidence for an early trapping of Env at HIV-1 assembly sites in the context of fully infectious virus assembly. This observation is consistent with the results of previous replication-incompetent HIV-1 model studies where Env appears to be directly trapped at the nascent Gag lattice sites via interactions between the MA domain of Gag and the C-terminal tail of Env. Furthermore, this study confirms the low number of Env molecules outside of the assembly site, supporting the proposed model of low Env incorporation into virus particles due to its intracellular retention.

## Figures and Tables

**Figure 1 viruses-13-00608-f001:**
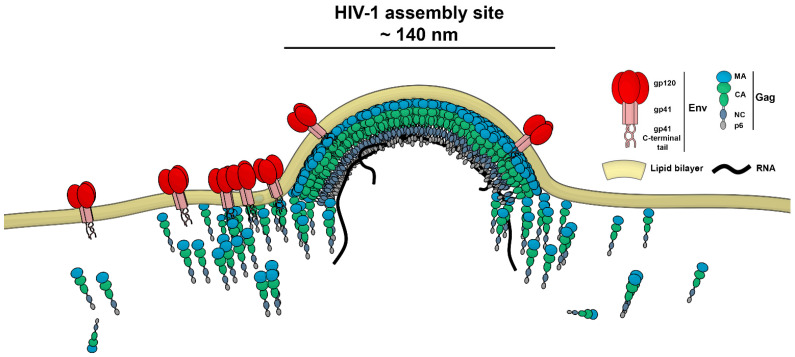
Schematic illustration of Gag and Env distribution during HIV-1 assembly on the cell plasma membrane.

**Figure 2 viruses-13-00608-f002:**
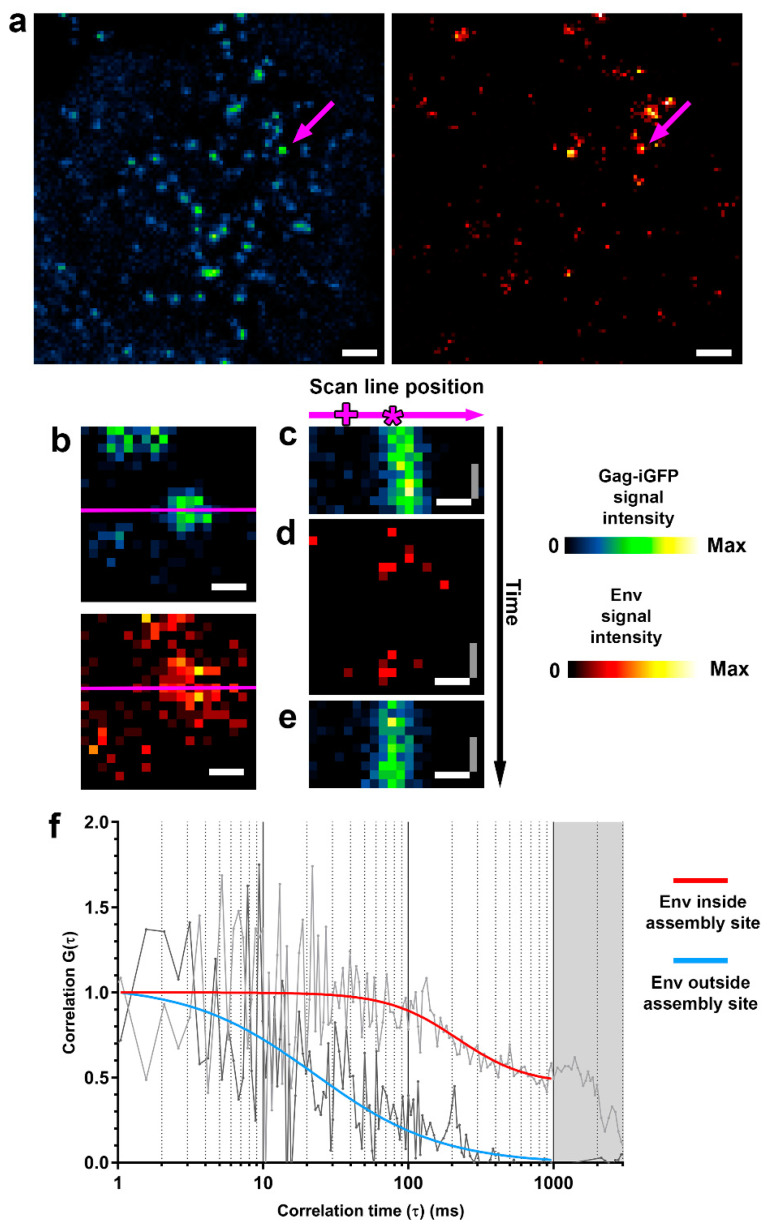
Two-colour sSTED-FCS allows for the discrimination of Env mobility inside and outside HIV-1 assembly sites. (**a**) Confocal overview images of live Jurkat T-cells infected with NL4.3 Gag.iGFP HIV-1. On day 3 post-infection, cells were adhered to poly-L-coated coverslips and monitored for the appearance of HIV-1 assembly sites prior to sSTED-FCS measurements. Image represents Gag.iGFP (blue-green) and Env (orange) signal at the cell-coverslip interface 20 min post-adherence. The arrows indicate an individual HIV-1 assembly site, subsequently analysed by sSTED-FCS. Scale bar: 1 μm. (**b**) Confocal close-up of an HIV-1 virus assembly site using Gag.iGFP signal (upper panel) as a guide and to align it with the position of the scanned line (magenta) across the Env signal (lower panel). Scale bar: 200 nm. (**c**–**e**) Representative intensity fluctuation carpets for Gag.iGFP signal in confocal mode prior to (**c**) and post (**e**) acquisition of the corresponding intensity fluctuation carpet for Env in STED mode (**d**) inside and outside the HIV-1 assembly site. Image x- and y-axis correspond to the position on the scan line and signal intensity at each time point, respectively. Scale bars: x-axis (white) = 200 nm, y-axis (grey) = 2.08 ms. (**f**) Representative normalised autocorrelation curves of Env diffusion from inside (light grey) and outside (dark grey) the virus assembly site obtained from a single position on the scan line within correlation carpets (marked by the asterisk (*) and plus sign (+), respectively, in **c**–**e**). Env signal autocorrelation curves for inside and outside virus assembly site (red and blue, respectively) were fitted using two-dimensional diffusion model. Greyed out area corresponds to the photobleaching-only portion of the correlation data.

**Figure 3 viruses-13-00608-f003:**
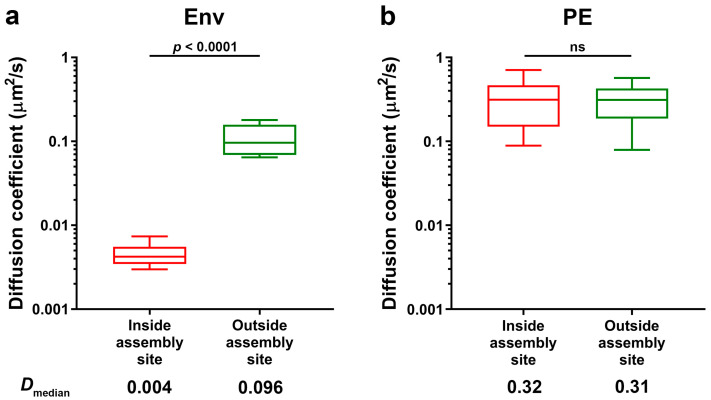
Env (**a**) and phosphatidylethanolamine (PE) (**b**) mobility inside and outside HIV-1 assembly sites of NL4.3 Gag.iGFP HIV-1 infected Jurkat T-cells. Median Env and phosphatidylethanolamine (PE) diffusion coefficients (*D_a_*) were determined by sSTED-FCS measurements of 20 sites each from two independent virus preparations and infections. Box and whisker plots (horizontal line—median, box—25–75% percentiles or interquartile range (IQR) and whiskers—10–90% measurements) showing *D_a_* values inside and outside assembly sites (red and green, respectively). Statistical significance was assessed by Wilcoxon rank-sum test.

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
