# Peer review of "Super-Resolution STED Microscopy-Based Mobility Studies of the Viral Env Protein at HIV-1 Assembly Sites of Fully Infected T-Cells"

_viruses, 2021, doi:10.3390/v13040608_

Round 1

Reviewer 1 Report

Chojnacki et al. analyzed the mobility of HIV-1 Env protein at the inside and outside of the virus assembly area by the novel sSTED-FCS method, and found that Env mobility is reduced at the assembly sites. This result is interesting, but the data is not enough to conclude so. The authors have previously reported that Env protein undergoes a virion maturation-induced increase in mobility using the same method (Nature Communications). In the previous study, they performed several control experiments using CT-truncated Env, GPI-SNAP, and MHC-I. However, in this study, they used only MHC-I as a control. Furthermore, based on the previous result, it is easy to speculate that Env mobility at assembly site is slow, because Gag protein is not processed yet at the assembly site. To make sure the conclusion, more control experiments are needed, for example CT-truncated HIV-1 Env, VSV-G (is incorporated into HIV-1 particles), and GALV Env (is incorporated into HIV-1 particles in the presence of Vpu, but not in the absence of Vpu).

Reviewer 2 Report

17 Mar 202

Review of:

Chojnacki and Eggeling

Super-resolution STED microscopy based mobility studies of the viral Env protein at HIV-1 assembly sites of fully infected T-cells

This is a very clear, well done study applying superresolution STED imaging combined with fluorescence correlation spectroscopy (FCS) to assess the dynamics of HIV assembly and budding from the plasma membrane of an infected cell. Specifically, by using time-resolved high resolution imaging with 100nm resolution, the study shows that Env diffusion is markedly reduced at sites of virus assembly.  This finding supports the current understanding that Gag complexes anchor at the plasma membrane, recruit Env molecules to the microdomain of virus assembly, and ‘trap’ Env molecules there.

I have no suggestions for improvement to the basic presentation of the methods and results, which is clear and well articulated.  There are a number of areas where the grammar and language are a bit challenging in English, or errors exist, and for which the authors could make some simple changes to make the manuscript more readable.  These are:

Line 43            “cells surface” should be “cell surface”

Line 62            “HIV-1 assembly site” should be “The HIV-1 assembly site”

Line 65            “between assembling” should be “between the assembling”

Line 67            “cell plasma membrane (MA) mediation of” should be “cell plasma membrane (MA), mediation of”

Line 72-73       “in the process regulated by interactions between MA domain of Gag and 72 Env C-terminal tail [12–14].”  should be “in a process regulated by interactions between the MA domain of Gag and the Env C-terminal tail [12–14].”

Line 179          “Jurakt” should be “Jurkat”

Line 181          “MA and CA domains” should be “the MA and CA domains”

Line 193          “(denoted intensity fluctuation carpet)” should be in quotes: “(denoted ‘intensity fluctuation carpet’)”

Line 194          “outside virus assembly site.” should be “outside the virus assembly site.”

Line 212          “outside HIV-1 assembly site.” should be “outside the HIV-1 assembly site.”

Line 224          “outside HIV-1 assembly site.” should be “outside the HIV-1 assembly site.”

Line 228          “from a single positions” should be “from a single position”

Line 244-246   For clarity, re-phrase this as: “the previously observed lower values of the Env diffusion coefficient in transfected HeLa cells was a result of performing this experiment with a STED observation spot of a smaller size – 55 nm instead of 100 nm in diameter, as in this study – which was only possible due to the high signal-to-noise ratio of the Env signal in the transfected cell model.”

Line 262          “utilising Gag model” should be “utilising a Gag model”

Line 267          “site” should be “sites”

Line 274-279   For clarity, re-phrase this as two sentences: “Unlike in camera-based methods like TIRFM [17], the presented approach does not allow for continuous monitoring of HIV-1 assembly sites over an extended period of time, due to movement of the cell and assembly sites. Nonetheless, the sSTED-FCS approach enables measurement of the average Env mobility in the time window between the initial individual Gag cluster formation and virus budding.”

Line 282          “has provided” should be “provide”

Line 286-287   “low number of Env molecules outside of the assembly site supporting the proposed model” should be “low numbers of Env molecules outside of the assembly site, supporting the proposed model”

Line 291          “was” should be “were”

Overall, this is a simple, elegant study that confirms current thinking about the dynamics of HIV assembly at the plasma membrane.

Round 2

Reviewer 1 Report

A control experiment that I suggested was added in the revised manuscript (Figure 3).